Increasing the explainability and success in classification: many-objective classification rule mining based on chaos integrated SPEA2

Yildirim Suna 1
Alatas Bilal balatas@firat.edu.tr 2
1 Data Processing Department, Secretary general of Special Provincial Administration , Elazig , Turkey
2 Department of Software Engineering, Firat (Euphrates) University , Elazig , Turkey
Sohaib Osama
Electronic publication date: 2024 Sep 6
Publication date: 2024
Volume: 10
Electronic Location ID: e2307
Received 2024 May 18; Accepted 2024 Aug 14
Copyright: ©2024 Yildirim and Alatas
Copyright year: 2024
Copyright holder: Yildirim and Alatas
License: This is an open access article distributed under the terms of the Creative Commons Attribution License, which permits unrestricted use, distribution, reproduction and adaptation in any medium and for any purpose provided that it is properly attributed. For attribution, the original author(s), title, publication source (PeerJ Computer Science) and either DOI or URL of the article must be cited.
License URL: https://creativecommons.org/licenses/by/4.0/

Keywords: Many-objective optimization, Interpretability, Machine learning, Metaheuristics, Rule mining

Funding: The authors received no funding for this work.

==============================
Classification rule mining represents a significant field of machine learning, facilitating informed decision-making through the extraction of meaningful rules from complex data. Many classification methods cannot simultaneously optimize both explainability and different performance metrics at the same time. Metaheuristic optimization-based solutions, inspired by natural phenomena, offer a potential paradigm shift in this field, enabling the development of interpretable and scalable classifiers. In contrast to classical methods, such rule extraction-based solutions are capable of classification by taking multiple purposes into consideration simultaneously. To the best of our knowledge, although there are limited studies on metaheuristic based classification, there is not any method that optimize more than three objectives while increasing the explainability and interpretability for classification task. In this study, data sets are treated as the search space and metaheuristics as the many-objective rule discovery strategy and study proposes a metaheuristic many-objective optimization-based rule extraction approach for the first time in the literature. Chaos theory is also integrated to the optimization method for performance increment and the proposed chaotic rule-based SPEA2 algorithm enables the simultaneous optimization of four different success metrics and automatic rule extraction. Another distinctive feature of the proposed algorithm is that, in contrast to classical random search methods, it can mitigate issues such as correlation and poor uniformity between candidate solutions through the use of a chaotic random search mechanism in the exploration and exploitation phases. The efficacy of the proposed method is evaluated using three distinct data sets, and its performance is demonstrated in comparison with other classical machine learning results.

Introduction

Classification rule mining represents a significant field of machine learning, with the objective of uncovering concealed patterns, dependencies, and relationships that underpin decision-making processes. Rule mining methodologies facilitate informed decision-making by extracting comprehensible rules from data that are challenging for humans to comprehend. Given the inherent complexity of contemporary data sets, there is a continual requirement for innovative, sophisticated, and adaptable methodologies within this branch of machine learning. Metaheuristic optimization-based solutions inspired by natural phenomena or social dynamics have succeeded in providing a paradigmatic change in this field. These solution approaches not only optimize accuracy but also enable the development of interpretable and scalable classifiers. Unlike heuristic methods or greedy search strategies, optimization-based rule inference approaches use mathematical optimization methods to systematically pursue the most discriminant rules. This allows optimizing performance metrics such as accuracy, precision and recall when working on high-dimensional, noisy and imbalanced data. Furthermore, these approaches facilitate the incorporation of domain-specific constraints and preferences, rendering them suitable for a multitude of real-world applications.

Glass-box type machine learning methods, which are based on metaheuristic optimization, provide interpretable information to the end user/decision maker. This information shows which attributes are taken into account and to what extent in the classification. In metaheuristic-based rule inference, the solutions found are expressed descriptively with “if-else” structures. In these methods, the relevant data set is considered as the search space and a random search is performed with a certain number of candidate solutions (population). Different metaheuristic algorithms, such as genetic algorithms (Martín, Rosete & Herrera, 2014), evolutionary algorithms (Anand, Vaid & Singh, 2009), and swarm optimization (Yildirim & Alatas, 2021; Barut & Yildirim, 2024), can be used for random search. In the iterative solution search process, candidate performances are evaluated according to a single or multiple (multi/many) conflicting objective functions. The method employed is referred to as multi-objective optimization when the number of objective functions is two or three, and many-objective optimization when there are more. As it is possible to negotiate complex trade-offs between conflicting objectives in multi/many optimization, it is possible to present multiple alternatives and balanced solutions to the decision maker. The advantages of multi/many optimization methods in rule inferences can only be seen by developing and adapting correct representation forms. It has been demonstrated that the utilization of appropriate methodologies can facilitate the generation of beneficial outcomes in a range of domains, including medical diagnosis (Habib et al., 2020), risk analysis (Soui et al., 2019), customer segmentation (Ncir, Qaffas & Bouaguel, 2023) and bioinformatics (Altay & Alatas, 2020). Conversely, in the existing literature, single or multi-objective methods have been employed for metaheuristic optimization-based rule extraction. To the best of the authors’ knowledge, however, a many-objective method has not yet been proposed.

This study examines the development of rule inference approaches based on metaheuristic many-objective optimization. To this end, Chaotic Rule Based- Strength Pareto Evolutionary Algorithm2 (CRb-SPEA2), which was developed by taking into account the advantages and disadvantages of existing methods, is introduced. Unlike single and multi-objective based rule extraction algorithms, the CRb-SPEA2 algorithm can perform automatic rule extraction by optimizing many more objectives simultaneously. The method, which employs the SPEA2 infrastructure, an evolutionary state-of-the-art metaheuristic algorithm, has been adapted for rule mining with a bespoke representation format that has been developed. Furthermore, it addresses issues such as correlation and poor uniformity between candidate solutions through the chaotic random search mechanism employed in the exploration and exploitation phases. To the best of our knowledge, the specificities of this method, which was proposed for the first time in the literature and whose performance was tested on different data sets, can be summarized as follows:

– This approach represents a novel application of the metaheuristic many-objective approach to rule mining.

– It is capable of performing automatic rule extraction by simultaneously optimizing four distinct performance metrics.

– A study area titled “Many-objective rule mining” has been proposed in the literature. Following the publication of this article, which represents a pioneering study in this field, it is anticipated that different methods will be proposed for this problem area.

– An interpretable artificial intelligence method has been proposed. It is expected that this method will lead to high-performance interpretable artificial intelligence studies.

– The method allows for the working on existing values without the need for discretization steps that are frequently used in other rule extraction methods.

– It uses a chaotic random search mechanism that can minimize weak statistical properties in the exploration and exploitation phases.

The following sections of the study are organized as follows: while literature summaries are presented in the Related Works section, basic information about many-objective optimization and the methodologies used are explained in detail in the Materials and Methods section. In the Result and Discussion sections, the data sets and experimental parameters used are explained, and the experimental results are shared. Finally, the article is concluded with the Conclusions section.

Related Works

Rule mining has been employed effectively to address numerous challenges for an extended period. For instance, they have been successful in identifying concealed patterns within expansive data sets and pattern discovery (Langhnoja, Barot & Mehta, 2013). Additionally, the discovery of such patterns plays a pivotal role in determining the features that exert the most significant influence on classification. Lutu & Engelbrecht (2010) demonstrated that more efficient models can be constructed through feature selection using rule mining. Rule mining is also an effective solution method in decision support systems and automatic decision-making mechanisms (Cheng et al., 2013; Hayes-Roth, 1985). The best examples of this in the literature can be seen in the business world and market analysis. While Kaur & Kang (2015) demonstrated how rule mining can contribute to the development of product recommendations and inventory management in retail and e-commerce, Krishan (2023) demonstrated that customer behaviour and demographic characteristics can also be examined with this method.

Rule mining is also employed successfully in the resolution of issues that arise in contemporary information technologies. Significant cyber security solutions can be provided by rule mining in the automatic detection of fraudulent transactions and behaviours that cannot be identified with conventional methods (Sarno et al., 2015). While Barut & Yildirim (2024) demonstrated that the minimum makespan value in cloud technologies can be optimized with rule-based methods, Sozou et al. (2017) emphasized the advantages of these methods in facilitating informatics-based scientific discoveries and generating hypotheses. Duch, Setiono & Zurada (2004) captured insights that were difficult to emerge immediately through traditional statistical analysis with rule mining and increased the depth of data understanding. This profound comprehension has facilitated the generation of bespoke, customer-centric solutions within the e-commerce domain (Okoye et al., 2013).

The machine learning methods employed in the literature for rule mining and the taxonomy of the method proposed in this study are illustrated in Fig. 1. The decision tree (Bashir et al., 2014) and Random Forest (Sirikulviriya & Sinthupinyo, 2011) based rule inference models are tree-based models that can create if-else based boards for classes with recursive mechanisms. Repeated Incremental Pruning to Produce Error Reduction (RIPPER) is a recursive induction algorithm that is specifically designed for rule-based classification (Ata & Yildiz, 2012). Bayes-based methods create classification rules based on probabilistic relationships between attributes and classes using Bayes’ theorem (Langseth & Nielsen, 2006). Association rule mining algorithms such as Apriori and FP-growth (Frequent Pattern-growth) arrive at classification rules by considering class label and attribute combinations together (Bala, KaramiLawal & Zakari, 2016). Metaheuristic optimization-based approaches, including this study, treat the data set as a search space and create rules in random search processes with the generated candidate solutions (Corcoran & Sen, 1994). The number of objectives targeted when creating the rule class is a crucial factor in determining the method to be employed.

Figure 1 Machine learning methods used in rule mining classification.

Al-Maqaleh (2021) put forth a methodology for identifying intriguing classification rules through an evolutionary metaheuristic approach. The research is concerned with the automatic discovery of augmented generation rules through the use of evolutionary algorithms. The study emphasizes the significance of accurate and engaging information for users, underscoring the necessity for high prediction accuracy and comprehensibility in classification rule discovery. Sağ & Kahramanlı Örnek (2022) present a novel classification rule mining (CRM) model based on Pareto-based multi-objective optimization, designated as CRM-PM, for datasets comprising multiple classes. In order to enhance the precision of classification, the proposed model initially treats the rule mining process as a constrained optimization problem and then transforms it into MOP (multi objective optimization problems) by leveraging the Pareto-dominance concept. This approach allows for the simultaneous optimization of two conflicting goals, which are significant challenges in rule mining. An approach for interpretable rule extraction in multi-objective metaheuristic data mining has been presented (Kalia et al., 2018). The study examined the challenges associated with handling high-dimensional problems, maintaining the comprehensibility of fuzzy rules, and optimizing multiple objectives in fuzzy rule-based systems. The issue is addressed through the use of a multi-objective genetic algorithm. The experimental results have demonstrated the potential for the design of understandable fuzzy rule-based classification systems for high-dimensional data classification problems. Rule inference conducted for a single purpose is referred to as single-objective (Corcoran & Sen, 1994; Yildirim, Yildirim & Alatas, 2021), whereas rule inference conducted for two or three purposes is designated as multi-objective (Yildirim & Alatas, 2021). This study introduces a novel approach to many-objective classification rule mining, which, to the best of the authors’ knowledge, has not been proposed previously.

In order to highlight the advantages of the proposed method, it is necessary to consider the general limitations of classical classification rule mining methods. One such limitation is human bias. The interpretation of rules can be affected by human bias, which leads to subjectivity in the rule inference process. The method proposed in this study can minimize this effect because it performs rule inference automatically and by taking into account trade-offs between multiple objectives. In contrast, classical methods can produce complex and less interpretable rules. The proposed method offers interpretable solutions based on if-else (glass-box) without causing any discretization loss. Additionally, classical methods may encounter problems in unbalanced data sets where one class significantly outnumbers the others, leading to the creation of rules that are biased against the majority class. The proposed method, on the other hand, is stronger against unbalanced data because it performs separate random search processes with a certain number of candidate solutions for each class. Furthermore, the proposed method shares some limitations with other methods. These include scalability for the data set size and number of features, overfitting leading to classification generalization, and parameter sensitivity resulting from the internal structure of the method used.

Material and Methods

This section will present the fundamental mechanisms of the proposed many-objective metaheuristic optimization-based rule extraction method and the specifics of the algorithms utilized. To this end, we will initially provide a synopsis of the essentials of multi/many-objective optimization, after which we will introduce SPEA2, the foundational algorithm of the proposed method. The details of the chaotic rule-based SPEA2 (CRb-SPEA2) algorithm developed for the proposed method and adapted for rule-based inference will be presented.

Multi/Many objective optimization

Multi/many objective optimization (MOO) deals with problem solutions that aim to optimize multiple conflicting goals simultaneously. The basic principle in MOO is to identify a solution set created by the trade-off between different targeted objectives. The number of conflicting objectives allows the relevant problem to be classified as multi- or many-objective. If the number of objectives is two or three, the problem is called multi-objective, and for more objectives, it is called many-objective. A plethora of methodologies have been proposed in the literature for the resolution of MOO problems (Stewart, Palmer & DuPont, 2021; Taha, 2020). Classical solution techniques, such as weighted sum, lexicographic and ɛ-Constraint, evaluate the objectives of the problem by placing them according to certain weights or in order. In these methodologies, which are referred to as prior approaches, decision makers are involved in the solution process at the outset. Decision makers can prioritize goals or reduce multiple goals to a single goal using certain ratios. Conversely, the involvement of decision makers in these methods introduces subjectivity and limitations to the exploration process. While the subjective preferences of the decision maker may result in suboptimal or biased solutions, these preferences may also lead to a narrow focus on a specific region of the search space. Another approach is the posteriori approach, also known as the Pareto-based approach. In such approaches, the objectives are treated equally, and a solution set is presented that includes non-dominated results that do not have absolute superiority over each other. The decision maker then evaluates this solution set and reaches a conclusion. These approaches are suitable for problems where the decision maker does not have to have information about the goals or where the desired goals are of equal importance. The non-domination criterion between candidates is of significant importance in the creation of the Pareto-front. In Pareto-based approaches, the domination criterion is of decisive importance. In a maximization problem such as the one presented in this study, the domination criterion is expressed as in Eq. (1). If this criterion is met, the solution vector s1 → is said to dominate the solution vector s2 →. In this context, the variable k represents the number of objectives, the variable i represents the relevant objective index in the solution vector, and the function f. represents the fitness function that determines the fitness value for the candidate. (1) ∀i∈1,2,3,…,k,Fs1i≥fs2i∩∃i∈1,2,3,…,k:fs1i>fs2i.

In addition to prior and posterior methods, there are also progressive methods in which the decision maker or designer can intervene throughout the current iteration and hybrid approaches where all these methods are used together (Luo et al., 2022). The choice of method is determined by the type of problem, the decision maker/designer and the importance of the targeted goals. However, in NP-hard (non deterministic polynomial time) optimization problems where deterministic solutions are inadequate, the increase in the number of objectives will be an important factor in determining the type of method to be chosen. In such complex problems, posteriori approaches may be preferred instead of priori approaches due to the subjectivity problem. In particular, posteriori methods using metaheuristic mechanisms are very effective in solving such NP-hard problems. Algorithms such as SPEA2 (Zitzler & Thiele, 1999; Zitzler, Laumanns & Thiele, 2001), NSGA-II/III (Non-dominated Sorting Genetic Algorithm-II/III) (Deb et al., 2002; Deb & Jain, 2014) and MOPSO (multi-objective particle swarm optimization) (Junjie et al., 2009) have been applied to a wide range of MOO problems, demonstrating their effectiveness in addressing the challenges posed by these complex problems.

In this study, the posteriori solution type was selected in order to exclude user influence in rule extraction. Furthermore, the extensive feature space employed and the high number of objectives led us to the conclusion that posteriori solutions utilizing metaheuristic mechanisms were the most appropriate. Literature review and previous experience of the authors show that SPEA2 algorithm has some advantages in this type of problems. For example, SPEA2 has better resource utilization cost since it does not perform multiple Pareto management like NSGA-II and III. It also provides a more successful diversity since it uses k-nearest based density estimation instead of the grid based distance criterion in MOGWO (Many Objective Grey Wolf Optimization) (Yildirim, 2022).

Strength Pareto Evolutionary Algorithm 2 (SPEA2)

The classical SPEA (Zitzler & Thiele, 1999) is a state-of-the-art MOO algorithm that identifies Pareto solution sets for problems containing multiple conflicting objectives. The classical SPEA, which employs evolutionary mechanisms, always considers the dominance and density factors. The dominance factor controls the superiority of the candidates over each other, while the density factor observes the distribution of the solutions found in the solution space and helps to create a balanced Pareto front. However, classical SPEA may require more generations to reach optimal solutions, which may result in slower convergence. Furthermore, the simple distance-based density estimation it employs may also diminish the efficacy of Pareto-front formation. Consequently, SPEA developers have proposed the SPEA2 algorithm, which represents an enhanced iteration of this algorithm (Zitzler, Laumanns & Thiele, 2001). While SPEA2 offers faster convergence than SPEA, utilizing elitism and tournament mechanisms, it also enables more successful Pareto-front formation with an advanced density estimation method.

SPEA2 has an initial population (P0) and an empty archive (P¯0) in the first iteration (t = 0). These two entities have a dynamic structure and are updated at each iteration according to dominance evaluations and algorithm mechanisms (Pt and P¯t). Generation control is carried out according to the fitness values of both population and archive individuals. The fact that individuals dominated by the same archive individuals have similar fitness values negatively affects density. To prevent this, SPEA-2 takes into account both the dominance and being dominated statuses of each candidate. The number of solutions dominated by the ith individual in Pt and P¯t is its Strength value (Si) and is calculated with Eq. (2). Here, the symbol . represents cardinality, the symbol +represents multiset union, and the symbol ≻ represents Pareto dominance. (2) Si=jj∈Pt+P¯t⋀i≻j.

The Si value is employed to calculate the raw fitness value, denoted by (Ri), which elucidates the dominance status of the ith individual. The greater the value of Ri determined by Eq. (3), the more the related candidate is dominated. If Ri isequal to zero, the candidate is considered to be non-dominated. (3) Ri= ∑j∈Pt+P¯t,j≻iSj.

The raw fitness value provides an indication of the dominance status of the candidate, but may be insufficient for accurate evaluation if the number of non-dominated candidates is high. Therefore, density information is also used in addition to the raw fitness value. The k -th nearest neighbour technique is used for density estimation (Silverman, 1986). Consequently, the intensity at a given point is a decreasing function of the distance to the k-th nearest data point. For each ith individual, the distances of all other individuals in the objective space to the j-th individual are calculated both in the archive and in the population and stored in a list. In the list organized in ascending order, the k-th value gives the desired distance and is denoted by σik. The k value depends on the population size (N) and the archive size (N¯) and is calculated with k=N+N¯. The density value of a candidate is inversely proportional to σik and is found by Eq. (4). Here, θ is a constant used to ensure that the denominator value is positive, and generally θ = 2. Consequently, a candidate’s fitness value (Fi isdetermined by Eq. (5) according to raw-fitness (Ri) and density values D(i)). (4) Di=1σik+θ

(5) Fi=Ri+Di.

The next generation is created (environmental selection) according to the Fi values of the candidates. Initially, non-dominated individuals with fitness values less than 1 are copied to the next generation’s archive, as illustrated in Eq. (6). During this process, the archive size of the next generation is taken into account. If P¯t+1=N¯, the environmental selection process is completed. In the case of P¯t+1<N¯, the previous archive and the best N¯−P¯t+1 non-dominated individual in the population are added to the next generation archive. In the case of P¯t+1>N¯, the candidate is removed from the next generation archive until P¯t+1=N¯ is achieved. (6) P¯t+1=ii∈Pt+P¯t⋀Fi<1.

The candidate extraction process considers the distance (σik) between individual i in set P¯t+1 and its k-th nearest neighbour. The individual with the shortest distance is selected, as demonstrated in Eq. (7). In the event that there are multiple candidates with the same distance, the equality is broken by taking the second smallest distances into account. (7) i≤dj⇔∀0<k<P¯t+1:σik=σjk∨∃0<k<P¯t+1:∀0<l<k:σil=σjl⋀σik<σjk.

The Simulated Binary Crossover (SBX) operator was employed as crossover operator (Deb & Agrawal, 1994). SBX generates two new individuals according to the probability distributions of the parent individuals (pr1, pr2). The polynomial probability distribution that ensures that the newly produced individuals are related to their parents is a function of the spread factor (β) and is expressed as in Eq. (8). The similarity of the new individuals to the parent individuals is determined by the constant φ (φ ∈ R+) . A high φ value increases the similarity of the new individuals to the parent individuals. The spread factor value for each variable h of the candidate solution vector is distinct (βh) and is determined by Eq. (9). Here, ϑc, represents the random constant between 0 and 1 generated for the variable h. At the conclusion of the crossover process, both newly produced individuals (sv1, sv2) are identified by Eqs. (10)–(11).

(8) Pβ=0.5φ+1βφ,ifβ≤1,0.5φ+11βφ+2,otherwise

(9) βh=2ϑc1φ+1,ifϑc≤0.5,121−ϑc1φ+1,otherwise

(10) sv1=0.51+βhpr1+1−βhpr2

(11) sv2=0.51−βhpr1+1+βhpr2.

The mutation operation is performed for all vector variables of the selected individual. The amount of change in the solution vector in each variable is calculated with (Δch) Eq. (12) and added to the relevant variable. The amount of change depends on the predefined mutation coefficient (γ) and a random number (ϑmh) generated for the variable h . (12) Δch=2ϑmh1γ+1−1,ifϑmh≤0.5,1−2ϑmh1γ+1,otherwise.

Many objective chaotic rule-based SPEA2 and evaluation principles

This study proposes a novel many-objective metaheuristic method for machine learning-based rule extraction. To this end, the SPEA2 algorithm has been adapted to address the rule inference problem. This necessitates two fundamental changes to the SPEA2 algorithm. The first is to design an appropriate representation form for candidate solutions. The second is to determine how to evaluate the objective values according to the rule compliance. In the metaheuristic-based rule extraction technique, the representation forms and search technique of the candidate solutions to be used are of significant importance. In the proposed method, the candidate solution employs a representation format comprising three distinct vectors. Consequently, candidate solution Vi comprises sub vectors V→ib,V→ilandV→iu (i = 1, 2, …, n). Vib=v1b,v2b,…,vjb,…,vnb is a binary vector, indicating which attributes will be utilized in the rule that the candidate will develop. If the value of vjb is greater than the predefined λ threshold value, as indicated in Eq. (13), the j-th attribute is included in the rule that the candidate will develop. (13) vj=1Featurejisusedintheruleifvib>λ,λϵ0,10otherwise.

Throughout the iterations, candidates obtain two values, one lower and one upper, for each attribute of the search space. A candidate keeps the upper/lower values found for each attribute in Vil=v1l,v2l,..vjl,..vnlandViu=v1u,v2u,..vju,..vnu vectors (vjl,vjuϵR). In an iteration, the upper and lower values found for the jth attribute must fall between the minimum (Ll and maximum (Uu values in the search space of the relevant attribute. Therefore, condition Ljl≤vjl<vju≤Uju must be constantly checked. At each iteration, candidates are presented with a single rule, which is updated throughout the process. The vjb value plays a pivotal role in the formation of these rules, while vjb and vjb provide context and meaning. To illustrate, consider a search for a class “C” that includes attributes 4., 7., and 10. in the ith iteration v4b,v7b,andv10b>λ. In this case, the candidate’s rule expression for the data set attributes (F1, F2, …, Fn) will be as in Eq. (14). This type of rule inference has two important contributions to increasing interpretability. The first is to show which attributes contribute to success and in what ranges. The second is that no discretization mechanism is needed to determine the lower and upper limits. Thus, there is no loss of information. (14) ifv4l≤F4≤v4uandv7l≤F7≤v7uandv10l≤F10≤v10uthenC.

All candidates assess the rules they have derived throughout the iterations in accordance with their performance in the search space. A candidate’s performance is contingent upon the degree of consistency exhibited by the rule they have derived for each class of data. In the evaluation of rules, the “if” and “then” components of the rule produced by the candidate solution Vi are taken into account. The attribute values and class of the data being compared determine which components of the rule are deemed to be consistent with them. Following the comparison in accordance with Table 1, the true positive (TP), true negative (TN), false positive (FP) and false negative (FN) metrics of the candidate solution are updated. These metrics are then employed in order to calculate the candidate’s goal values.

Table 1 Calculation of TP, TN, FP and FN values.

“if” part	“then” part	Operation	
True False	True False	increase TP by 1 increase TN by 1	
True	False	increase FP by 1	
False	True	increase FN by 1	

In multi-objective optimization methods, the contradiction between the selected objectives is of significant importance. The trade-offs resulting from this contradiction help to form the Pareto-front. Pareto solutions assist decision makers in interpreting trade-offs between objectives. Given that this study is a data mining optimization problem, it is of paramount importance to select the most appropriate metrics. In data mining, precision and recall are metrics that may conflict with one another. Precision represents the ratio of true positive predictors to all positive predictions. Recall is defined as the ratio of true positive predictions among all actual positive predictions. Consequently, in certain instances, an increase in one of these metrics may result in a decrease in the other. Two additional metrics that may conflict are accuracy and F1-score. Accuracy is a measure of the overall accuracy of the model’s predictions, whereas F1-score represents the harmonic average of the precision and recall metrics. The conflict between these two metrics is particularly evident in unbalanced data sets. In this case, it would be misleading to consider only the accuracy metric. On the other hand, since F1-score combines precision and recall into a single metric, the trade-off between these two metrics cannot be seen. In data mining, it is possible to present Pareto solutions that show all trade-offs between these metrics to the decision maker with many-objective optimization. Consequently, in this study, Pareto fronts were constructed on the basis of the trade-offs between these four metrics.

The objective values for a candidate solution are calculated with the previously obtained total TP, TN, FP and FN values at the end of each iteration. Equations (15)–(18) illustrate the calculation of the Accuracy (Acc), Precision (Pre), Recall (Rec) and F1-score (F1) objectives, respectively. The i-th candidate solution maintains these four metrics in the objective vector Os→=Acc,Pre,Rec,F1. The dominance relationship between candidates is determined by comparing these objective vectors

(15) Acc=TP+TNTP+TN+FP+FN

(16) Pre=TPTP+FP

(17) Recall=TPTP+FN

(18) F1=2∗Pre∗RecPre+Rec.

The randomness mechanism is of paramount importance in the exploration and exploitation processes of metaheuristic methods. Some traditional random number generation functions may exhibit statistical weaknesses, such as poor distribution properties, insufficient uniformity, or correlations between generated values. These weaknesses can lead to biased results, especially in simulations and modelling. In order to address this issue, researchers have examined chaotic maps due to their deterministic behaviour, longer periods and advanced statistical properties. In Yildirim et al. (2021), the authors of this paper tested the performance of chaotic functions in metaheuristic approaches. As a result of the study, it was observed that the tent function can be partially more resistant to local minima (or maxima). For this reason, tent function is also preferred in random process management in this study. It is well documented in the literature that metaheuristic algorithms employing chaotic maps yield competitive outcomes and offer flexibility, particularly during the exploration phase. This study opted to utilize chaotic maps in the randomness mechanism of SPEA2 due to the flexibility they afford, especially during the exploration process. Given the promising results demonstrated by the authors in their previous studies, the tent chaotic map (Li et al., 2017), whose mathematical expression is given in Eq. (19), was employed. This adapted version was designated as chaotic SPEA2 (CRb-SPEA2). (19) Xn+1=∂Xn,Xn<12∂1−Xn,12≤Xn.

The chaotic map is dependent on the real ∂ constant and initial (X0) values. By setting these two coefficients correctly, the sequence produced exhibits chaotic behaviour. In CRb-SPEA2, the V→ib,V→ilandV→iu initial vectors of the candidate solution, ϑc and ϑmh values in crossover and mutation operations are generated by the Tent chaotic map function. While the pseudo code demonstrating all these processes of the rule extraction method with CRb-SPEA2 is presented in Algorithm 1 , the basic mechanisms of the algorithm are shown in Fig. 2.

Figure 2 The basic mechanism of CRb-SPEA2.

Algorithm-1. The pseudo-code of the proposed CRb-SPEA2	
1.     Define all constant and pre-defined parameters;N,N¯,γ, φ, ∂,λ	
2.     InitializePtandPt, t = 0(andcreateV→ib,V→ilandV→iuforViusing the Tent chaotic map)	
3.     Create the rules for each individuals (byEq.13-14)	
4.     While(t <maximum iteration number)	
5.     Calculate fitness of individuals according to their rule evaluation (byTable 1,Eq.15-18thenEq.1-5)	
6.     Copy all nondominated individuals inPtandPt, toPt + 1	
7.     Ifsize ofPt + 1exceedsNthenreducePt + 1(byEq.7)	
8.     Else ifsize ofPt + 1is lessN¯thenthen fillPt + 1with dominated individuals	
9.     Evolutionary Operations (byEq.8-12)	
10.     Update the individuals	
11.     t = t + 1	
12.     End While	
	

Results and Discussion

The performance of CRb-SPEA2 has been observed for three different data sets, two of which are well known in the literature and one of which is obtained from a real engineering problem. The details of these data sets, which are both balanced and unbalanced, are provided in Table 2. The Ecoli data set (Nakai, 0000), which contains 336 samples from these data sets, contains information about the protein localization sites in Escherichia coli bacteria. The RAC dataset is the second dataset and was used to determine the concrete component amounts as a result of the data obtained from the debris as a result of the devastating earthquake that occurred in Elazig (Turkey) in 2020 (Ulucan et al., 2023). The Iris data set contains class information for three different flower species. 10 independent experiments were performed for all data sets. In the study, the statistical results of independent experiments performed for each data set, the rules drawn, and the training and test results of some experiments are presented separately to give an idea. In order to facilitate a comparison of the metric performances, the results of well-known state-of-the-art machine learning algorithms, namely naive Bayes (NB), k-nearest neighbor (kNN), support vector machine (SVM), decision tree (DT), Multiobjective Evolutionary Fuzzy Classifier (EFC), JRIP, Ridor, Random Forest (RF), Hyperpipes (HP), AdaboostM1 (AB) were utilized. All experimental parameters are presented in Table 3. Given that the proposed system is a supervised machine learning method, the data sets were divided into training and test data sets. The performance of the rules derived from the training data sets was evaluated separately with the test data, and the results of both phases were shared. In the experiments, the CRb-SPEA2 algorithm was written and tested in Python. The results of other machine learning algorithms were obtained from WEKA software.

Table 2 Characteristics of the data sets used.

Dataset	Field	Tot.samples	Classes	Samples	Features	Min.val.	Max.val.	
Ecoli	Bioinformatics	336	cp	143	mcg	0	0.89	
im	77	gvh	0.16	1	
pp	52	lip	0.48	1	
imu	35	chg	0.5	1	
om	20	aac	0	0.88	
omL	5	alm1	0.03	1	
imL	2	alm2	0	0.99	
imS	2				
RAC	Civil Engineering				rac	320	420	
Class-A	41	water	96	210	
Class-B	42	fine	481	679	
Class-C	37	coarse1	521	1367	
		coarse2	0	865	
Iris	Life	150			sepal_length	4.3	7.9	
Setosa	50	sepal_width	2.0	4.4	
Versicolor	50	petal_length	1.0	6.9	
Virginica	50	petal_width	0.1	2.5	

Table 3 Experimental parameters.

Parameters	Value	
Independent experiments for each data set	10	
Split ratio for Training-Test data sets (%)	70-30	
N	20	
N¯	20	
λ	0.5	
γ	5	
φ	5	
∂	0,2∈R	
Crossover Probability	0.8	
Mutation Probability	0.1	
Max. Iteration	500	

The initial experiments were conducted on classes belonging to the Ecoli data set. The statistical results for the accuracy (Acc), precision (Pre), recall (Rec) and F1 metric values of the Pareto candidates obtained in the training and test phase experiments are presented in Tables 4 and 5. Figure 3 illustrates the sample 4D distribution (the fourth dimension is represented by color) of the training Pareto candidates that produce high-performance rules. Table 6 presents a selection of example rules that were automatically derived for the relevant classes by Pareto candidates in the training experiments. As it is known, in Pareto-based multi/many objective methods, multiple solutions are presented to the decision maker instead of a single solution. For this reason, in Table 7 and subsequent comparison tables, some of the CRb-SPEA2 based solutions will be presented.

Table 4 Statistical results of the training stage of the Ecoli data set.

 	cp	im	imS	imL	imU	om	omL	pp	
 	Acc	Pre	Rec	F1	Acc	Pre	Rec	F1	Acc	Pre	Rec	F1	Acc	Pre	Rec	F1	Acc	Pre	Rec	F1	Acc	Pre	Rec	F1	Acc	Pre	Rec	F1	Acc	Pre	Rec	F1	
Averag e	0.783	0.799	0.758	0.746	0.780	0.703	0.532	0.503	0.730	0.024	0.588	0.043	0.910	0.087	0.769	0.137	0.789	0.430	0.603	0.423	0.801	0.366	0.661	0.372	0.865	0.301	0.823	0.345	0.820	0.586	0.748	0.591	
Media n	0.808	0.811	0.787	0.762	0.823	0.699	0.536	0.537	0.804	0.007	1.000	0.014	0.936	0.034	1.000	0.066	0.863	0.390	0.708	0.468	0.893	0.322	0.714	0.366	0.957	0.150	1.000	0.260	0.876	0.596	0.805	0.608	
Min	0.429	0.429	0.118	0.212	0.221	0.221	0.018	0.035	0.000	0.000	0.000	0.000	0.553	0.000	0.000	0.000	0.000	0.000	0.000	0.000	0.408	0.000	0.000	0.000	0.340	0.018	0.333	0.037	0.306	0.180	0.083	0.153	
Max	0.965	1.000	1.000	0.960	0.906	1.000	1.000	0.803	0.991	0.200	1.000	0.333	0.995	0.500	1.000	0.666	0.931	1.000	1.000	0.612	0.982	1.000	1.000	0.833	0.995	1.000	1.000	0.857	0.980	1.000	1.000	0.825	
Std	0.102	0.152	0.186	0.122	0.147	0.193	0.302	0.202	0.316	0.051	0.492	0.086	0.116	0.136	0.421	0.185	0.200	0.213	0.284	0.139	0.179	0.279	0.263	0.172	0.185	0.324	0.202	0.268	0.139	0.220	0.221	0.149	

Table 5 Statistical results of the test stage of the Ecoli data set.

 	cp	im	imS	imL	imU	om	omL	pp	
 	Acc	Pre	Rec	F1	Acc	Pre	Rec	F1	Acc	Pre	Rec	F1	Acc	Pre	Rec	F1	Acc	Pre	Rec	F1	Acc	Pre	Rec	F1	Acc	Pre	Rec	F1	Acc	Pre	Rec	F1	
Average	0.772	0.752	0.781	0.738	0.737	0.626	0.533	0.490	0.610	0.004	0.176	0.008	0.892	0.112	0.231	0.132	0.758	0.335	0.474	0.320	0.806	0.381	0.670	0.376	0.874	0.265	0.647	0.302	0.795	0.560	0.627	0.517	
Median	0.797	0.795	0.833	0.756	0.841	0.666	0.636	0.597	0.762	0.000	0.000	0.000	0.930	0.000	0.000	0.000	0.821	0.296	0.454	0.312	0.886	0.250	0.666	0.348	0.940	0.105	1.000	0.190	0.841	0.500	0.687	0.538	
Min	0.415	0.415	0.190	0.320	0.010	0.003	0.010	0.010	0.400	0.000	0.000	0.000	0.544	0.000	0.000	0.000	0.000	0.000	0.000	0.000	0.386	0.000	0.000	0.000	0.346	0.000	0.000	0.000	0.247	0.173	0.062	0.111	
Max	0.900	1.000	1.000	0.886	0.910	1.000	1.000	0.806	0.980	0.040	1.000	0.076	1.000	1.000	1.000	1.000	0.930	1.000	1.000	0.640	0.980	1.000	1.000	0.800	1.000	1.000	1.000	1.000	0.930	1.000	1.000	0.774	
Std	0.103	0.140	0.181	0.111	0.240	0.274	0.329	0.252	0.375	0.011	0.381	0.020	0.123	0.272	0.421	0.287	0.197	0.252	0.287	0.167	0.182	0.312	0.294	0.181	0.168	0.324	0.412	0.302	0.145	0.234	0.221	0.151	

Figure 3 Sample Pareto solutions obtained for each class of the Ecoli data set at the end of the training phase.

Table 6 Some example rules obtained for the classes of the Ecoli data set.

Acc	Pre	Rec	F1	RULES			
0.963	0.960	0.950	0.955	if 0.152 <mcg <0.705 and 0.198 <gvh <0.588 and 0.150 <alm1 <0.560 then cp		
0.891	0.897	0.833	0.864	if 0.148 <mcg <0.696 and 0.160 <gvh <0.503 and 0.123 <alm1 <0.557 then cp		
0.900	0.800	0.727	0.761	if 0.013 <mcg <0.711 and 0.668 <alm1 <0.940 then im		
0.841	0.600	0.818	0.692	if 0.063 <mcg <0.849 and 0.646 <alm1 <0.928 then im		
0.930	1.000	0.363	0.533	if 0.760 <mcg <0.880 then imU		
0.881	0.470	0.727	0.571	if 0.451 <mcg <0.880 and 0.184 <alm1 <0.815 and 0.666 <alm2 <0.979 then imU		
0.970	1.000	0.500	0.666	if 0.754 <aac <0.880 then om		
0.980	1.000	0.666	0.800	if 0.672 <aac <0.880 then om		
1.000	1.000	1.000	1.000	if 0.504 <lip <1.000 and 0.500 <chg <0.500 and 0.005 <alm2 <0.699 then omL		
0.910	0.888	0.500	0.640	if 0.584 <gvh <0.877 and 0.500 <chg <0.500 and 0.278 <aac <0.552 and 0.037 <alm1 <0.729 then pp		
0.910	0.888	0.500	0.640	if 0.560 <gvh <0.871 and 0.500 <chg <0.500 and 0.263 <aac <0.555 and 0.030 <alm1 <0.732 then pp		

Table 7 Comparative results of CRb-SPEA-2 (test stage) and well-known ML algorithms.

	cp	im	pp	imU	om	omL	
	Acc	Pre	Rec	F1	Acc	Pre	Rec	F1	Acc	Pre	Rec	F1	Acc	Pre	Rec	F1	Acc	Pre	Rec	F1	Acc	Pre	Rec	F1	
CRb-SPEA2	0.910	0.844	0.970	0.903	0.910	0.800	0.726	0.761	0.920	0.900	0.562	0.692	0.932	1	0.361	0.533	0.982	1	0.667	0.800	1	1	1	1	
0.960	0.960	0.950	0.955	0.811	1	0.136	0.24	0.912	0.889	0.500	0.640	0.881	0.470	0.727	0.571	0.972	1	0.499	0.444					
0.923	0.873	0.960	0.915	0.544	0.323	1	0.488	0.693	0.333	0.937	0.491	0.910	0.571	0.727	0.640	0.603	0.130	1	0.230					
0.948	0.978	0.900	0.938	0.871	0.736	0.636	0.682	0.910	0.705	0.750	0.727	0.871	0.450	0.818	0.580									
0.868	0.773	0.980	0.864	0.861	0.75	0.545	0.631	0.554	0.254	0.937	0.400	0.920	1	0.272	0.428									
NB	0.841	0.944	0.919	0.893	0.841	0.810	0.810	0.810	0.841	0.810	0.850	0.829	0.841	0.714	0.833	0.769	0.841	1	0.750	0.87	0.841	1	0.500	0.667	
kNN	0.811	0.878	0.973	0.923	0.811	0.708	0.810	0.756	0.811	0.875	0.700	0.778	0.811	0.636	0.583	0.609	0.811	0.857	0.750	0.800	0.811	1	1	1	
DT	0.811	0.854	0.946	0.897	0.811	0.640	0.762	0.696	0.811	0.889	0.800	0.842	0.811	0.750	0.500	0.600	0.811	1	0.875	0.933	0.811	1	1	1	
SVM	0.811	0.949	1	0.974	0.811	0.618	1	0.764	0.811	0.850	0.850	0.850	0.811	–	0	0	0.811	1	0.625	0.769	0.811	0.667	1	0.800	
EFC	0.710	0.878	0.977	0.925	0.710	0.529	0.818	0.643	0.710	0.621	0.857	0.720	0.710	–	0	–	0.710	–	0	–	0.710	1	1	1	
JRIP	0.833	0.896	0.977	0.935	0.833	0.792	0.864	0.826	0.833	0.882	0.714	0.789	0.833	0.813	0.813	0.813	0.833	0.556	0.625	0.588	0.833	–	0	–	
RIDOR	0.824	0.955	0.955	0.955	0.824	0.633	0.864	0.731	0.824	0.850	0.810	0.829	0.824	0.769	0.625	0.690	0.824	0.857	0.750	0.800	0.824	–	0	–	
RF	0.859	0.956	0.977	0.966	0.859	0.700	0.955	0.808	0.859	0.818	0.857	0.837	0.859	0.900	0.563	0.692	0.859	1	0.750	0.857	0.859	1	0.500	0.667	
HP	0.640	0.755	0.909	0.825	0.640	0.432	0.864	0.576	0.640	0.909	0.476	0.625	0.640	0.333	0.063	0.105	0.640	1	0.375	0.545	0.640	–	0	–	
AB	0.578	0.611	1	0.759	0.578	0.524	1	0.688	0.578	–	0	–	0.578	–	0	–	0.578	–	0	–	0.578	–	0	–	

The experimental results are discussed throughout the article, while the metric representations of the candidates are presented in [Acc Pre Rec F1] format. The large number of examples of a class in the data set increases the rule diversity, due to the greater diversity of examples available. In order to give an idea to the readers, sample Pareto solution distributions obtained from the training phases of some data sets are given (Figs. 3, 4 and 5). In these distributions, the trade-offs between the solutions, the number of solutions and the diversity in the solution space can be better seen. As illustrated in Fig. 3, distinct rule sets can be generated for the decision-maker by Pareto candidates in classes such as cp, iml and pp, which have a relatively large number of examples. In contrast, at most one or two rules are derived for classes such as omL, iml and imS, which contain a small number of data. In the Ecoli data set, CRb-SPEA2 was able to generate Pareto candidates with performance ranging between 0.900 and 1.00 in almost all classes and all metrics. These results were also observed in the test data, with the highest objective metrics obtained from class rules such as omL which have a small number of data. This is to be expected, given the nature of these classes. The limited data set resulted in wide rule intervals, which increased the rule compliance of the data set. In classes containing a relatively large amount of data, the maximum Acc, Pre and Rec metric performances generally remained above 0.900 during the training and testing phases. When compared to the results obtained by well-known ML algorithms for the same classes, CRb-SPEA2 succeeded in producing non-dominated solutions for all classes. At the same time, the highest Acc value in all classes was obtained by CRb-SPEA2. Indeed, it has achieved considerable success in certain classes. To illustrate, for the cp class [0.960, 0.960, 0.950, 0.955], the Pareto candidate demonstrated superior performance compared to other ML algorithms, except for SVM. In SVM, a non-dominance result emerged due to the high Recall and F1 metrics. In the omL class, which contains a limited number of data points, CRb-SPEA2 was able to infer a rule that could encompass all the data and outperformed the metrics of other ML algorithms. Upon examination of the sample rules generated by Pareto candidates, it was observed that mcg plays a pivotal role in the cp, im and imU classes, while gvh and alm1 also exhibited notable efficacy. In the om and omL classes, mcg has no effect, whereas chg, gvh and alm1 are determinants.

The RAC dataset was used to determine the amounts of concrete components from the classified construction and demolition wastes generated after the Sivrice-Elaz ığ (Turkey) earthquake and to estimate the early age concrete strength class. The statistical outcomes of the training and testing phases with CRb-SPEA2 are presented in Tables 8 and 9. The sample Pareto candidate distribution obtained during the training phase is illustrated in Fig. 4. Upon examination of the training results, it was observed that the highest accuracy value for Pareto candidates was 0.761, 0.738 and 0.904 for Class A, B and C, respectively. In the independent experiments, in the training phases, the Pareto curve usually consisted of three or four candidates and high diversity was observed. Although the Precision and Recall metrics reached 1.00 in different classes, the trade-offs between them were high. The prominent solutions were [0.771, 0.750, 0.750, 0.750, 0.750] for Class-A, [0.778, 1.00, 0.214, 0.353] for Class-B and [0.905, 0.833, 0.625, 0.714] for Class-C. In general, the best results were obtained for class C, and the accuracy values for this class ranged from 0.600 to 0.910. This was also the case for Precision, while Recall did not achieve very high results. As illustrated in Tables 10 and 11, when compared to the results of other ML algorithms, the CRb-SPEA2 test results still managed to produce non-dominated values in both classes. However, the highest accuracy values were obtained by classical ML algorithms, with the NB, DT, RIDOR and EFC algorithms achieving particularly outstanding scores. Upon examination of the sample rules automatically generated by Pareto candidates during the training phase, it is seen that cement and water properties play an important role in Class-A and Class-B, while in Class-C, in addition to these, the coarse2 attribute also emerges as a dominant factor.

Figure 4 Sample Pareto solutions obtained for each class of the RAC data set at the end of the training phase.

Figure 5 Sample Pareto solutions obtained for each class of the Iris data set at the end of the training phase.

The final experiments were conducted on the Iris data set (Fisher, 2021), one of the most well-known data sets in the literature. The statistical results of the training and test experiments performed with this data set are presented in Tables 12 and 13. In all three classes of this data set, CRb-SPEA2 achieved results above 0.950 in all metrics. As illustrated in Fig. 5, the density of the Pareto fronts is relatively low during the training process, indicating that the density tracking function is effective in this data set, as in other data sets of SPEA2. This performance of Pareto solutions in the training experiments was maintained in the testing stages, and even some rule inferences were obtained in the Setosa class to cover all test data. Upon examination of the results presented in Table 14, it can be seen that CRb-SPEA2 has been able to produce experimental results that are superior to those of other classical machine learning algorithms in the Setosa class. Non-dominated solutions were produced for the Virginica and Versicolor classes. It can be observed that the majority of methods achieved similar results in this data set, where classification performance was high. The most significant advantage of CRb-SPEA2 is that it offers interpretable rule sets that reveal this performance. The interpretable sample rules automatically produced by Pareto solutions in Table 15 demonstrate that the petal-length parameter is sufficient for the Setosa and Virginica classifications. In addition, the sepal-length and sepal-width attributes can be used for the Versicolor classification.

Table 8 Statistical results of the training stage of the RAC data set.

 	Class-A	Class-B	Class-C	
 	Acc	Pre	Rec	F1	Acc	Pre	Rec	F1	Acc	Pre	Rec	F1	
Average	0.624	0.667	0.633	0.592	0.666	0.650	0.393	0.428	0.791	0.620	0.531	0.530	
Median	0.630	0.720	0.600	0.595	0.666	0.589	0.393	0.439	0.797	0.583	0.563	0.495	
Min	0.476	0.476	0.330	0.428	0.595	0.421	0.214	0.353	0.666	0.312	0.375	0.417	
Max	0.761	0.750	1.000	0.750	0.738	1.000	0.571	0.480	0.904	1.000	0.625	0.714	
Std	0.117	0.130	0.302	0.138	0.058	0.243	0.146	0.053	0.097	0.283	0.106	0.126	

Table 9 Statistical results of the test stage of the RAC data set.

 	Class-A	Class-B	Class-C	
 	Acc	Pre	Rec	F1	Acc	Pre	Rec	F1	Acc	Pre	Rec	F1	
Average	0.666	0.666	0.671	0.656	0.676	0.767	0.327	0.397	0.776	0.474	0.740	0.526	
Median	0.666	0.714	0.750	0.642	0.706	0.813	0.286	0.402	0.777	0.417	0.667	0.508	
Min	0.547	0.516	0.450	0.515	0.555	0.444	0.166	0.285	0.644	0.231	0.625	0.375	
Max	0.761	0.750	0.850	0.750	0.738	1.000	0.571	0.500	0.904	0.833	1.000	0.714	
Std	0.096	0.104	0.163	0.098	0.079	0.231	0.170	0.088	0.106	0.252	0.168	0.140	

Table 10 Some example rules obtained for the classes of the RAC data set.

Acc	Pre	Rec	F1	RULES	
0.761	0.750	0.750	0.750	if 320.0 ≤ cement ≤ 387.201 and 113.247 ≤ water ≤ 195.021 then Class-A	
0.738	1	0.214	0.353	if 320.0 ≤ cement ≤ 398.133 and 96.0 ≤ water ≤113.204 then Class-B	
0.904	0.833	0.625	0.714	if 391.205 ≤ cement ≤ 420.0 and 99.622 ≤ water ≤154.391 and 0.0 ≤ coarse2 ≤ 865.02 then Class-C	

Table 11 Comparative results of CRb-SPEA2 (test stage) and well-known ML algorithms.

	Class-A	Class-B	Class-C	
	Acc	Pre	Rec	F1	Acc	Pre	Rec	F1	Acc	Pre	Rec	F1	
CRb-SPEA2	0.771	0.750	0.750	0.750	0.778	1	0.214	0.353	0.771	0.833	0.625	0.714	
0.500	0.500	1	0.667	0.738	1	0.214	0.352	0.905	0.833	0.625	0.714	
0.571	1	0.100	0.181	0.667	0.500	0.500	0.500	0.889	1	0.333	0.500	
0.620	0.642	0.450	0.529	0.714	0.667	0.286	0.4	0.714	0.333	0.500	0.400	
0.571	0.667	0.200	0.307	0.714	1	0.143	0.250	0.833	0.500	1	0.666	
NB	0.888	0.889	0.889	0.889	0.888	0.833	0.833	0.833	0.888	1	1	1	
kNN	0.611	0.667	0.444	0.533	0.611	0.444	0.667	0.533	0.611	1	1	1	
DT	0.833	1	0.667	0.800	0.833	0.667	1	0.800	0.833	1	1	1	
SVM	0.722	0.818	0.950	0.883	0.722	0.571	0.667	0.615	0.722	–	0	–	
EFC	0.833	1	0.889	0.941	0.833	0.667	1	0.800	0.833	0.333	0.500	1	
JRIP	0.777	0.778	0.778	0.778	0.777	0.667	0.667	0.667	0.777	1	1	1	
RIDOR	0.833	1	0.889	0.941	0.833	0.800	0.667	0.727	0.833	0.600	1	0.750	
RF	0.722	1	0.444	0.615	0.722	0.545	1	0.706	0.722	1	1	1	
HP	0.666	0.800	0.889	0.842	0.666	0.500	0.667	0.571	0.666	–	0	–	
AB	0.611	0.800	0.889	0.842	0.611	–	0	–	0.611	0.375	1	0.545	

Table 12 Statistical results of the training stage of the Iris data set.

 	setosa	versicolor	virginica	
 	Acc	Pre	Rec	F1	Acc	Pre	Rec	F1	Acc	Pre	Rec	F1	
Average	0.801	0.740	0.844	0.756	0.926	0.912	0.880	0.880	0.863	0.852	0.764	0.778	
Median	0.828	0.747	0.900	0.754	0.938	0.941	0.914	0.908	0.871	0.914	0.828	0.779	
Min	0.352	0.339	0.514	0.507	0.733	0.744	0.200	0.333	0.723	0.548	0.257	0.409	
Max	0.990	1.000	1.000	0.985	0.980	1.000	1.000	0.972	0.961	1.000	1.000	0.941	
Std	0.142	0.197	0.150	0.111	0.046	0.075	0.161	0.107	0.071	0.135	0.203	0.135	

Table 13 Statistical results of the test stage of the Iris data set.

 	setosa	versicolor	virginica	
 	Acc	Pre	Rec	F1	Acc	Pre	Rec	F1	Acc	Pre	Rec	F1	
Average	0.849	0.802	0.920	0.830	0.923	0.928	0.852	0.862	0.851	0.823	0.766	0.759	
Median	0.900	0.913	1.000	0.840	0.955	0.933	0.933	0.928	0.866	0.882	0.800	0.817	
Min	0.333	0.333	0.660	0.500	0.688	0.789	0.066	0.125	0.688	0.517	0.200	0.315	
Max	1.000	1.000	1.000	1.000	0.977	1.000	1.000	0.967	0.977	1.000	1.000	0.965	
Std	0.160	0.214	0.106	0.132	0.060	0.071	0.216	0.156	0.098	0.165	0.245	0.193	

Table 14 Comparative results of CRb-SPEA-2 (test stage) and well-known ML algorithms.

	Setosa	Versicolor	Virginica	
	Acc	Pre	Rec	F1	Acc	Pre	Rec	F1	Acc	Pre	Rec	F1	
CRb-SPEA2	1	1	1	1	0.977	0.937	1	0.967	0.977	1	0.933	0.965	
0.977	0.937	1	0.967	0.955	1	0.866	0.928	0.933	1	0.8	0.888	
0.955	1	0.866	0.928	0.933	0.833	1	0.909	0.955	0.882	1	0.937	
0.866	0.714	1	0.833	0.955	0.933	0.933	0.933	0.955	0.933	0.933	0.933	
0.844	0.618	1	0.810	0.911	0.789	1	0.882	0.911	1	0.733	0.846	
NB	0.955	1	1	1	0.955	0.889	1	0.941	0.955	1	0.867	0.929	
kNN	0.955	1	1	1	0.955	0.889	1	0.941	0.955	1	0.867	0.929	
DT	0.955	1	1	1	0.955	0.889	1	0.941	0.955	1	0.867	0.929	
SVM	0.955	1	1	1	0.955	0.889	1	0.941	0.955	1	0.867	0.929	
EFC	0.980	1	1	1	0.955	0.889	1	0.974	0.955	1	0.867	0.970	
JRIP	0.901	1	0.933	0.966	0.901	0.850	0.895	0.872	0.901	0.882	0.882	0.882	
RIDOR	0.960	1	1	1	0.960	0.905	1	0.950	0.960	1	0.882	0.938	
RF	0.960	1	1	1	0.960	0.905	1	0.950	1	0.882	0.938	0.960	
HP	0.882	1	1	1	0.882	0.882	0.789	0.833	0.882	0.789	0.882	0.833	
AB	0.960	1	1	1	0.960	0.905	1	0.950	0.960	1	0.882	0.938	

Table 15 Some example rules obtained for the classes of the Iris data set.

Acc	Pre	Rec	F1	RULES	
1.000	1.000	1.000	1.000	if 1.167 <petal_length <2.944 then setosa	
0.977	0.937	1.000	0.967	if 1.076 <petal_length <3.633 then setosa	
0.955	0.933	0.933	0.933	if 0.841 <petal_width <1.645 then versicolor	
0.933	1.000	0.800	0.888	if 4.893 <sepal_length <6.895 and 2.000 <sepal_width <3.502 and 2.965 <petal_length <4.732 then versicolor	
0.955	0.933	0.933	0.933	if 4.801 <petal_length <6.900 then virginica	
0.977	1.000	0.933	0.965	if 4.988 <petal_length <6.900 then virginica	

Conclusions

This study demonstrates the applicability of metaheuristic many-objective optimization methods in interpretable rule inference-based classification problems. The CRb-SPEA2 algorithm, adapted to data mining problems, was employed in three independent experiments with three different benchmark data sets. The obtained results were also compared with those of the classical ML algorithms and the competitive non-dominated scores of the proposed method were demonstrated in all experiments. The interpretable rules derived by Pareto candidates during the training phase enable the decision maker to identify which attributes of the data set are decisive in the classification. Due to the nature of many-objective optimization methods, CRb-SPEA2 has been able to offer different rule sets with different trade-offs to the decision maker. Although high performance and explainability can be achieved, a limitation of many-objective optimization algorithms is that they are sensitive to their parameter settings and finding the right parameters can sometimes be time consuming. However, the many-objective optimization based rule mining method proposed in this paper is thought to help to create simpler and more interpretable and understandable rules by striking a balance between simplicity and complexity of rule sets. In this way, fewer and more descriptive rules can make it easier for users to understand and apply these rules. In the future, the authors aim to propose new adaptive and hybrid versions of the intelligent optimization methods for high performance rule mining problems. In addition, in order to reduce the model complexity and computational cost of this proposed method, parallel and distributed versions will be studied in order to obtain more efficient results on large data. The first applications of this method in fuzzy rule mining, association rule mining, and sequential pattern discovery are also aimed. The authors’ next work is to demonstrate that metaheuristic many-objective optimization methods can be applied to different engineering problems.

Supplemental Information

Supplemental Information 1 Source code of the proposed method

Supplemental Information 2 3 datasets

This study is derived from the PhD study titled “Development of Rule Mining Based Classification Models for Quantitative Data with Many-Objective Intelligent Metaheuristic Optimization Model”.

Additional Information and Declarations

Competing Interests

Author Contributions

Data Availability

Bilal Alatas is an Academic Editor for PeerJ.

Suna Yildirim conceived and designed the experiments, performed the experiments, analyzed the data, performed the computation work, prepared figures and/or tables, authored or reviewed drafts of the article, and approved the final draft.

Bilal Alatas conceived and designed the experiments, analyzed the data, authored or reviewed drafts of the article, and approved the final draft.

The following information was supplied regarding data availability:

The source codes and the datasets are available in the Supplemental Files.

The data sets are also publicly available at UCI Machine Learning Repository:

- Ecoli Dataset Online Address: http://archive.ics.uci.edu/dataset/39/ecoli

- Bupa Dataset Online Address: http://archive.ics.uci.edu/dataset/60/liver+disorders

- Iris Dataset Online Address: http://archive.ics.uci.edu/dataset/53/iris.

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
