# Peer review of "Increasing the explainability and success in classification: many-objective classification rule mining based on chaos integrated SPEA2"

_PeerJ Computer Science, doi:10.7717/peerj-cs.2307_

## Round 0.1 · original submission · Major Revisions

Please see both reviewers' detailed comments. The reviews commend its structure and writing but suggest several improvements: verify the applicability of optimization methods, explain the choice of SPEA-2 and Tent Map, add section numbers, provide software details, define all acronyms and symbols, discuss fuzzy classification applicability, enhance comments on figures, and clarify parameter definitions and multiple metric values. They also recommend discussing limitations, correcting table alignments, and expanding future research directions.

Reviewer 1 ·

Basic reporting

The authors in this paper attempted to optimize more than three objectives while increasing the explainability and interpretability of the classification task. The idea presented in this paper is good however, the authors are suggested to consider the following suggestions and comments.

1: The title talks about "Increasing the explainability" and the manuscript also discusses in the abstract on "interpretability" and other places.
The paper demonstrates classification performance without thoroughly addressing how the approach enhances explainability and interpretability. Only the rules are included in a few tables but not discussed.
How explainability and interpretability are increased in this paper? how the rules were derived or how they contribute to interpretability? How the rules can be used for decision-making, their advantages over traditional methods, and any trade-offs involved?

2: The claim that "there are limited studies on metaheuristic-based classification, and none optimize more than three objectives while enhancing explainability and interpretability" is a strong assertion. To make the claim more accurate and less absolute, it could be rephrased to reflect the extent of the authors' review without making an overgeneralized assertion such as "To the best of our knowledge, there are limited studies on metaheuristic-based classification that optimize more than three objectives while also focusing on enhancing explainability and interpretability."
Make the changes accordingly at all places.

3: Related Work: The authors are suggested to extend the literature and include and review recent studies.

Experimental design

4: Materials and Methods: The authors are suggested to include a pictorial representation of proposed chaotic rule-based SPEA-2 (CRb-SPEA-2) at start of this section that also helps to visualize the overall flow of steps and components in proposed approach and can be aided by the respective sections.

5: The authors are suggested to evaluate the performance and benchmark the proposed algorithm by using larger, more complex, and diverse datasets. Given the small size and traditional nature of the datasets used in this study seems less ideal for evaluating the performance and benchmarking of new algorithms.

6: The authors mentioned that "The method is applicable to both balanced and unbalanced datasets". However, for unbalanced datasets, this approach is not evaluated in this study. The authors are suggested to reevaluate the characteristics of the datasets and can include something like "Although not tested in this study, the method is designed to be applicable to both balanced and unbalanced datasets."

7: Line 453: "data sets were divided into training and test data sets". What is the split size for test and train dataset?

Validity of the findings

8: The authors are suggested to compare and discuss the performance of the proposed approach and results in the light of existing literature. Literature has a huge number of studies that have used these datasets for the classification task and reported the performance of different algorithms.

9: Improve the conclusion section and also highlight the limitations of this study and potential future directions.

Additional comments

10: Keeping in view the above comments, revise the contributions listed from Line 115 to 130.

·

Basic reporting

The authors propose a new methodology for the problem of rule mining task in explainable artificial intelligence studies. The proposed intelligent optimization based method is capable of performing automatic rule extraction by simultaneously optimizing four distinct success metrics (Accuracy, Precision, Recall, and F1). "Many-objective rule mining" has been proposed in the literature and this seems important for leading to high-performance explainable and interpretable artificial intelligence studies. The paper is well-structured and well-written. My suggestions on additions and modifications are listed below:

Experimental design

1. Has the existing literature verified the applicability of single-objective and multi-objective optimization methods in this many-objective optimization context? Are there previous studies demonstrating the benefits of the suggested approach?
2. Why SPEA-2 is adapted many-objective rule mining problem is not mentioned.
3. The paper structure should be corrected according to its Sections. There are not section numbers in the manuscript and organization of the paper at the end of Introduction section should be corrected depending on this.

Validity of the findings

4. The motivation on adding chaos into SPEA-2 is absent. Why specifically Tent Map among many other discrete and continuous chaotic maps is integrated into proposed optimization method is not clear. How the initial value of this map is selected is not mentioned.
5. Please add information about the software used for numerical simulation and optimization results.
6. All acronyms & symbols should be defined at their first use in the body text, even they are well known in literature.
7. Can this proposed methodology be used for fuzzy classification rule mining problem?
8. Figure 2 that provides an intuitive understanding of the algorithm’s tolerance is important. However, this needs a more explicit comment by the authors. These are also true for Figure 3 and Figure 4.

Additional comments

9. All of the parameters in the presented equations should be clearly defined and explained.
10. In the experimental results, it should be mentioned how the target explainability is increased.
11. Table 14 should be correctly aligned.
12. In Table 7, Table 11, and Table 14; the authors should clarify why there are more than one values for Acc, Pre, Rec, and F1 metrics in CRb-SPEA2 (This is due to many-objective approach however it will be better to mention).
13. Possible disadvantages and limitations of this many-objective rule mining methodology in interpretable and explainable artificial intelligence researches should be mentioned.
14. “The Simulated Binary Crossover (SBX) operator was employed as Crossover operator” in Line 315 should be corrected as “The Simulated Binary Crossover (SBX) operator was employed as crossover operator”. A reference should also be added for this special cross-over type.
15. Future research directions seem insufficient. The efficient usage of the proposed method in association rule mining, sequential pattern mining, fuzzy rule mining, etc. can be added. The potential research directions on performance increment of the proposed method can also be mentioned in the Conclusion section.

---

## Round 0.2 · accepted · Accept

I confirm that the authors have addressed all of the reviewers' comments.

Reviewer 1 ·

Basic reporting

Thank you for making an effort to address the comments.

Experimental design

no comment

Validity of the findings

no comment

·

Basic reporting

No Comment

Experimental design

No Comment

Validity of the findings

No comment